# Distal Fibula Reconstruction in Primary Malignant Tumours

Adyb Adrian Khal [1,2,*], Riccardo Zucchini [3], Claudio Giannini [3], Andrea Sambri [4,5], Davide Maria Donati [3,4] and Massimiliano De Paolis [5]

1   Department of Orthopaedics and Traumatology, Iuliu Hatieganu University of Medicine and Pharmacy, 400000 Cluj-Napoca, Romania
2   Department of Orthopaedics, Lenval University Children's Hospital, 06200 Nice, France
3   Department of Orthopaedic Oncology, Istituto Ortopedico Rizzoli, 40136 Bologna, Italy; riccardo.zucchini@studio.unibo.it (R.Z.); claudio.giannini@ior.it (C.G.); davidemaria.donati@ior.it (D.M.D.)
4   Department of Biomedical and Neuromotor Sciences (DIBINEM), University of Bologna, 40126 Bologna, Italy; andrea_sambri@libero.it
5   Department of Orthopaedics, IRCCS Azienda Ospedaliera Universitaria di Bologna, 40138 Bologna, Italy; massimiliano.depaolis@aosp.bo.it
*   Correspondence: khal.adyb@umfcluj.ro or adyb_khal@yahoo.com or khal.a@pediatrie-chulenval-nice.fr

**Abstract:** (1) Background: Restoration of ankle biomechanics after distal fibula (DF) resection in bone sarcomas can be performed with different techniques. We report the functional and oncological outcomes of a case series; (2) Methods: Ten patients (5 females and 5 males) with a mean age of 27 years (range 10–71) were retrospectively evaluated. Following the resection, different techniques were used to reconstruct the ankle: tibiotalar arthrodesis, residual lateral malleolus fixed to the tibia, non-vascularized or rotational vascularized fibula transposition and intercalary allograft. All complications were recorded, and the functional outcomes were evaluated; (3) Results: The mean follow-up time was 54 months (range, 13–116). Six patients were free of disease while four patients died of disease. All patients had a stable ankle and bone union, which was achieved after a mean of 9.4 months (range 3–20). The mean MSTS Score was 26.7 (range 21–30). Chronic ankle pain and peroneal external nerve palsy were observed. Patients underwent additional surgeries for deep infection and for equinus ankle deformity. No local recurrence was observed. Metastasis occurred in four patients after a mean of 14.7 months (range 2–34); (4) Conclusions: After DF resection, the restoration of ankle biomechanics gives acceptable functional results, but a larger series of patients with long-time follow-up are required to confirm the durability of the reconstruction.

**Keywords:** distal fibula; reconstruction; sarcoma

## 1. Introduction

Primary malignant bone tumours of the fibula are very rare. They account for approximately 3.2–5.6% of all primary bone sarcomas [1,2]. The distal fibula (DF) is less frequently involved (8–18% of cases) than the proximal one [1–3]. Also, sarcomas originating in the distal third of the fibula have a better prognosis than in other parts of the same bone [3]. Ewing sarcoma, osteosarcomas and fibrosarcomas are the most frequent malignant bone lesions that occur [1,3].

Due to the advances in chemotherapy, radiation therapy and surgical techniques, limb salvage has become the preferred treatment when possible. Moreover, limb salvage surgery is also possible in most of the cases as the fibula is an expandable bone. The fibula is frequently sacrificed when it is harvested as a non-vascularized or vascularized graft to reconstruct the bone defect after intercalary resections in tumoral cases, in septic or nonseptic non-unions, in adult traumatology or in congenital pseudarthrosis.

Given the rarity of primary bone sarcomas in this anatomical area, controversies exist regarding the optimal surgical treatment. The DF, together with the distal tibia and the talar dome, compose the ankle joint. During normal walking it supports five times the body's

weight, while during running the bear load is up to 13 times the body's weight [4]. The resection of the DF can alter foot and ankle biomechanics, thus generating ankle instability and/or ankle valgus deformity [5,6]. The aim of the surgery is to achieve negative resection margins with a good functional outcome. Ankle function can be generally preserved when the resection is at least 15–20 mm above the distal tibiofibular articulation [7] or 5 mm above the growth plate in skeletally immature patients [8]. If this is not possible because of oncological reasons, reconstruction is generally recommended.

Capanna [7] described in 1986, a series of 11 patients who underwent distal fibula resections and reconstruction for both benign and malignant tumours. Several possibilities to reconstruct the bone defect in order to restore the ankle function were reported. Later on, several techniques have been described in the literature, including reconstructions with vascularized [9,10] or non-vascularized autografts [11], allografts [12], ankle arthrodesis [11,13] and prosthesis [14].

All these methods have different advantages and disadvantages. In distal fibular resections without reconstruction, the loss of the stabilizing effect of the lateral malleolus is a challenge to be overcome [5,15]. The soft tissue reinforcement will not fully compensate and the ankle may be destabilized in valgus [5]. Proximal non-vascularized or vascularized fibular transposition may alter the knee's stability or endanger the peroneal external nerve [11]. Ankle arthrodesis is well tolerated, but the range of motion is lost.

Ideally, a perfect reconstruction technique should be easily reproducible, give a satisfactory functional outcome with no complications and be cost-effective.

Currently, only case reports and small case series have been described in literature. To the best of our knowledge, only one publication studied the follow-up, complications and functional outcomes of a larger group of patients (11 patients), but the main reconstruction technique used was ankle arthrodesis [11].

The aims of this study were to describe our experience in the treatment of primary malignant bone tumours of the DF according to different types of reconstruction, to analyse the complications and the functional outcomes and to determine whether the outcomes of patients undergoing reconstruction are comparable to the previously reported series in literature.

## 2. Materials and Methods

Ten patients (5 females and 5 males) affected by a primary malignant bone sarcoma of the DF who underwent surgery at a single institution were included (Table 1). The mean age at the time of surgery was 27 years (range, 10–71). The inclusion criteria for enrolling patients were as follows: patients with distal fibula primary malignant bone sarcomas, patients who underwent surgical treatment in a single institution between February 2001 and February 2018, patients whose clinical data, radiological data and histopathological data were complete, patients with at least 12 months of follow-up and patients who provided informed consent. This is an observational and descriptive study.

**Table 1.** Demographics and clinical data for patients included in the study.

| Patient # | Age at Surgery and Sex | Diagnosis | Tibial Lateral Cortex Involvement | Enneking Stage | Chemotherapy | Local Postoperative Radiation Therapy | Follow-Up Time (Months) | Oncological Status |
|---|---|---|---|---|---|---|---|---|
| 1 | 71 F | Parosteal Osteosarcoma | Yes | 1B | No | No | 23 | NED |
| 2 | 47 M | Osteosarcoma | Yes | 2B | Yes | No | 26 | DOD |
| 3 | 19 M | Ewing Sarcoma | Yes | 2B | Yes | Yes | 71 | DOD |
| 4 | 27 F | Ewing Sarcoma | Yes | 3 | Yes | Yes | 111 | NED |
| 5 | 19 F | Ewing Sarcoma | Yes | 2B | Yes | No | 18 | NED |
| 6 | 12 M | Ewing Sarcoma | No | 2B | Yes | Yes | 13 | DOD |
| 7 | 37 F | Parosteal Osteosarcoma | No | 1B | No | No | 48 | NED |
| 8 | 14 F | Ewing Sarcoma | No | 3 | Yes | No | 116 | NED |
| 9 | 17 M | Ewing Sarcoma | No | 2B | Yes | Yes | 44 | DOD |
| 10 | 10 M | Ewing Sarcoma | No | 2B | Yes | No | 69 | NED |

NED = No evidence of disease, DOD = Died of disease.

Patients were addressed to our institution because of swelling or leg pain with or without limping unrelated to previous injuries. The average duration of symptoms prior to diagnosis was 4.8 months (range 3–7).

All patients were pre-operatively assessed with radiographs, magnetic resonance imaging (MRI) and computerized tomography (CT) scans of the distal leg and ankle.

The use of radiotherapy and chemotherapy was decided at the discretion of a multidisciplinary team (orthopaedic surgeon, radiotherapist and medical oncologist). In the case of osteosarcoma, patients received chemotherapy according to the EURAMOS protocol [16] and, in the case of Ewing Sarcoma, patients received one of the following chemotherapy protocols according to the time frame: EW-NEO1, EW-NEO2, EW-NEO3, EW-pilot ISG, EW ISG/SSG3, ISG-AEIOP [17]. Eight patients were treated with neoadjuvant chemotherapy.

An antibiotic prophylaxis (amoxicillin/clavulanic acid 2200 mg plus gentamycin 240 mg) was given one hour prior to surgery. Additional doses of amoxicillin/clavulanic acid 1200 mg every eight hours plus gentamycin 240 mg were administered once a day for two days after surgery.

In all patients, the surgical procedure was performed under general anaesthesia, in a supine position and with a tourniquet applied at the proximal thigh. A direct lateral approach to the DF was performed. After incision, the dissection continued anteriorly towards the anterior extensor muscular lodge of the calf with identification of the tibia, interosseous membrane and the anterior tibial vascular pedicle and posteriorly, towards the posterior flexors muscular lodge with identification of the peroneal muscles and posterior tibial vascular pedicle. The fibular vascular pedicle was ligated proximal to the tumour and neurolysis of the external peroneal nerve was performed.

All resected specimens were analysed for surgical margins according to Enneking criteria [18]. The resection margins were considered intralesional if the dissection passed within the lesion; marginal if the dissection passed within the pseudocapsule or the reactive tissue around the lesion; wide if the lesion, the pseudocapsule or the reactive tissue and a surrounding cuff of normal tissue was removed en bloc and radical if the lesion, the pseudocapsule or the reactive tissue and the entire muscle or bone involved were removed as one block [18].

In the present series, different techniques have been used to reconstruct the bone defect. Allografts were taken from the local bone bank and stored according to standard musculoskeletal banking rules [19].

### 2.1. Tibiotalar Arthrodesis

Three patients (#1, #2 and #3) received reconstruction with ankle (tibiotalar) arthrodesis (AA). The foot was fixed in 5 degrees of dorsiflexion and 5 degrees of valgus. The arthrodesis was fixed with 2 screws and was reinforced with a fibula allograft (Figure 1A,B) (Patient #1) or with a fibula autograft (fibula diaphysis proximal to the tumour) in Patient #2. In patient #3, the AA was reinforced with both a fibular allograft and autograft (Figure 2A–C).

### 2.2. Rotational Vascularized Fibula Transposition

In one patient (#4), after the DF resection, the articular part of the lateral malleolus was preserved. The residual fibula was cut at the proximal metaphysis, distally to the insertion of the lateral collateral ligament of the knee. After the isolation and preservation of the fibular vascular pedicle, the graft was rotated 180 degrees, the proximal osteotomy of the graft being in contact with the residual peroneal malleolus and fixed to the tibia with screws (Figure 3A–C).

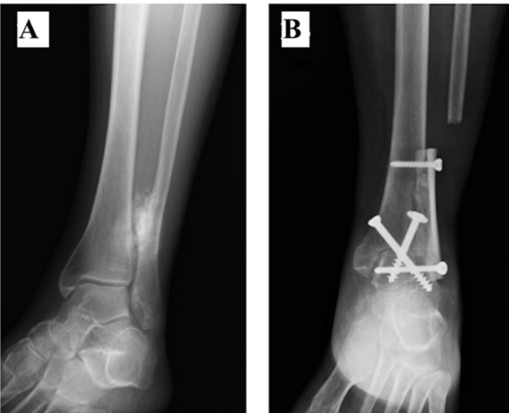

**Figure 1.** (**A**) Ankle external oblique Radiography showing a parosteal osteosarcoma of the distal fibula in a 71-year-old female. (**B**) After resection of 7.5 cm of the distal fibula and the lateral cortex of the tibia, the arthrodesis was fixed with 2 screws and was reinforced with a fibula allograft. The ankle AP Radiography shows the tibiotalar arthrodesis at 3 months of follow-up.

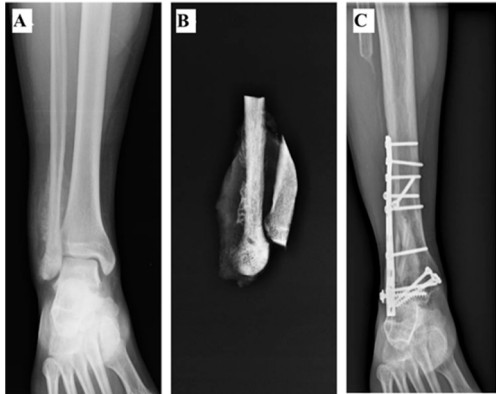

**Figure 2.** (**A**) Ankle AP Radiography showing a conventional osteosarcoma of the distal fibula in a 47-year-old male. (**B**) A radiography was performed for the resected specimen which consisted of 11 cm of the distal fibula and lateral cortex of the tibia. (**C**) Patient received a reconstruction with a tibiotalar arthrodesis which was fixed with 2 screws and was reinforced with a fibula autograft (fibula diaphysis proximal to the tumour) and a fibula autograft. The reconstruction was still stable at 5 years of follow-up.

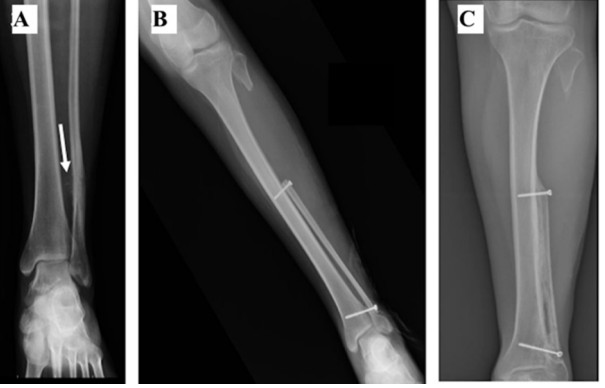

**Figure 3.** (**A**) Ankle AP Radiographs showing a Ewing Sarcoma of the distal fibula in a 27-year-old female. The lesion showed periosteal reaction and soft tissue invasion (arrow) and was located above the tibiofibular syndesmosis. (**B**) Following the tumour resection, the residual peroneal malleolus was 2 cm long and for the reconstruction a 180 degrees rotational fibula transposition was used. (**C**) Bone radiographic union was achieved after 20 months.

## 2.3. Non-Vascularized Fibula Transposition

In 2 patients (#5 and #6), after the DF resection (Figure 4A–D), the proximal part of the ipsilateral fibula was osteotomized at the proximal metaphysis level and the peroneal vessels were ligated. The graft was transposed distally in order to substitute the lateral malleolus and fixed to the tibia.

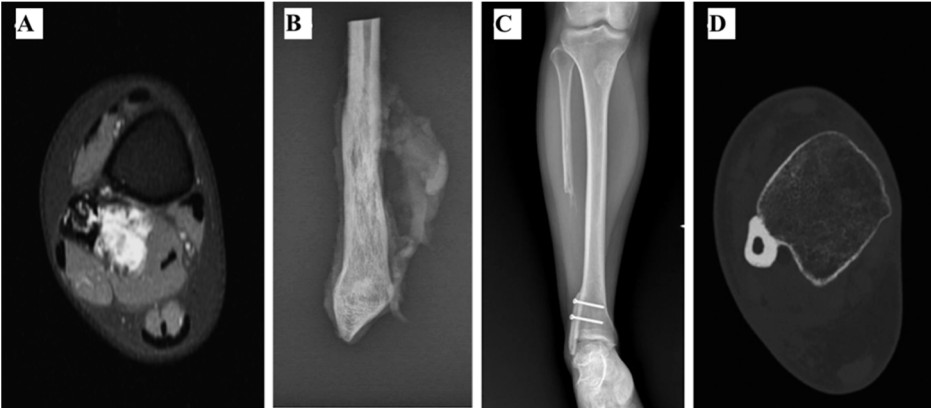

**Figure 4.** (**A**) Axial T1 MRI showing a Ewing Sarcoma of the distal fibula with a large posterior soft tissue invasion (hypersignal) in a 19-year-old female. (**B**) Radiography shows the resected distal fibula was 11.1 cm long. (**C**,**D**) Reconstruction of the lateral ankle was performed with a non-vascularized autograft. This was harvested from the fibular diaphysis proximal to the tumour and transposed distally to substitute the lateral malleolus and fixed to the tibia. Radiography (**C**) and Axial CT-Scan (**D**) show that bone union was achieved in 12 months after surgery.

## 2.4. Intercalary Allograft

In Patient #7, the bone defect was reconstructed with a 4 cm frozen fibula allograft fixed to the distal tibia (Figure 5A–D). The lateral malleolus was preserved.

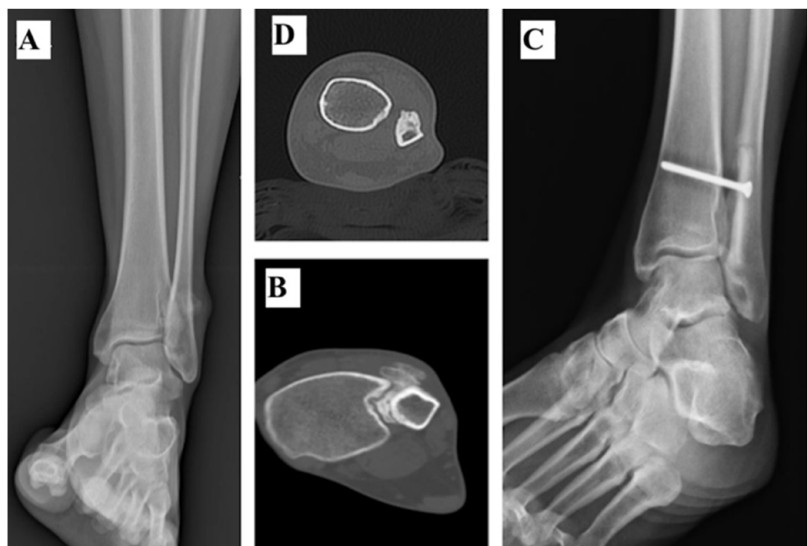

**Figure 5.** (**A**,**B**) AP ankle radiographs (**A**) and Axial CT-Scan (**B**) showing a parosteal osteosarcoma of the distal fibula in a 37-year-old female. (**C**,**D**) After the intercalary resection, a 4 cm allograft strut was used to reconstruct the bone defect and fixed to the tibia with a screw. At the last follow-up (48 months), the ankle external oblique Radiography (**C**) and the Axial CT-Scan (**D**) demonstrate the reconstruction was still stable, with a perfect integration without any sign of mechanical failure of the graft.

### 2.5. No Reconstruction

In 3 patients (#8, #9 and #10) the peroneal malleolus was preserved and the residual DF was fixed to the tibia with screws (Figure 6A,B).

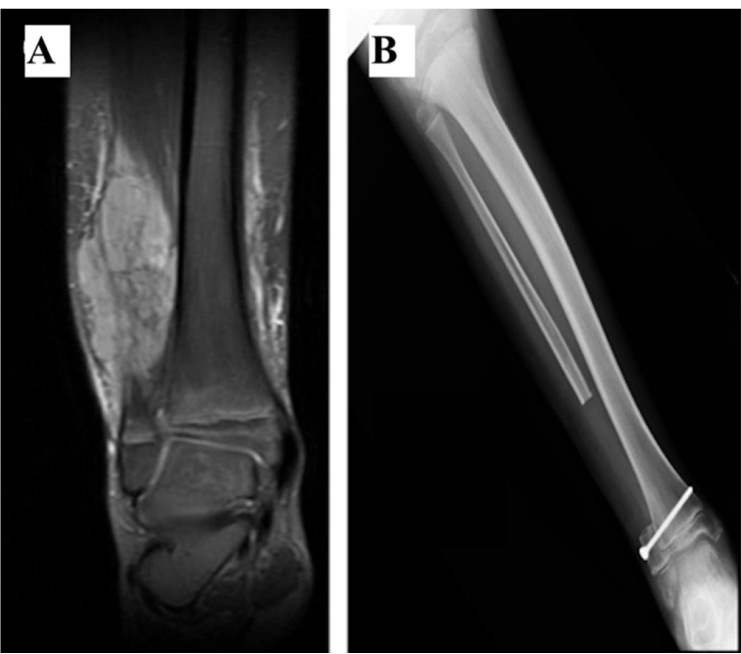

**Figure 6.** (**A**) Coronal Fat-Sat MRI showing a large tumour mass (Ewing Sarcoma) emerging from the distal fibula in a 10-year-old boy. (**B**) Following the resection, as the peroneal malleolus remained in site, reconstruction was not performed and a screw was fixed to stabilize the ankle.

After surgery, all patients had an immobilisation cast or splint for 4 weeks. No weight bearing was allowed during the first month after surgery, then partial weight bearing was allowed up to 12 weeks. Ankle mobilization was encouraged after cast removal in order to prevent ankle stiffness or muscle weakness.

Four patients underwent local postoperative radiation therapy. This was given concurrently with chemotherapy, 6 to 8 weeks after surgery in the case of intralesional or marginal resection of radio-sensitive tumours. The dose of radiation therapy was 54 Gy.

All complications were recorded. The clinical examination included the assessment of the stability of the ankle and the ankle range of motion (ROM). Ankle valgus deformity was measured on plain lower limb radiographs. The MSTS Score [20] was used to evaluate the functional outcome at the last follow-up.

The MSTS Score for the inferior limb is a rating scale that was first described by Enneking [20] in 1993 to assess the functional outcome postoperatively. It mainly includes 6 items: pain, function, emotional acceptance, support, walking limitation and gait, which are ranged between 0 and 5. The score has a maximum of 30 or if converted to percentage, 100. A total score < 15(50%) is considered poor, 15(50%)–17(59%) fair, 18(60%)–20(69%) moderate, 21(70%)–22(74%) good and 23(75%)–30(100%) excellent.

A descriptive study is presented and data are presented in total frequencies and percentages.

## 3. Results

### 3.1. Follow-Up

Patients were evaluated every 3 months during the first 2 years, every 6 months up to the 5th year after surgery, and once a year until 10 years after the surgery. The mean follow-up time was 54 months (range, 13–116). Six patients were free of disease at the last follow-up, while four patients died of disease.

### 3.2. Surgical Findings

Mean surgical time was 113 min (range 73–180 min). (Table 2).

Mean resection length was 116 mm (range 40–180 mm).

Lateral cortex of the distal tibia was also involved in five patients.

Surgical margins were wide in six patients and marginal in four patients.

### 3.3. Clinical and Radiological Findings and Functional Outcome

At the final follow-up, all living patients had a stable ankle at the clinical and radiological examination.

Bone union was achieved in all patients who underwent a reconstruction after a mean of 9.4 months (range 3–20).

The mean MSTS Score was 26.7 (range 21–30) (Table 2).

### 3.4. Complications

In one patient (#2) a deep infection was observed 49 days after surgery and required two surgical debridements and antibiotic therapy based on an antibacterial susceptibility test.

Peroneal external nerve palsy was observed in two cases (#4 and #8). One patient recovered nerve function after 4 months, whereas patient #8 had a permanent drop foot successfully managed with an ankle-foot-orthosis.

Patient #10 underwent an Achilles tendon lengthening for equinus ankle deformity 11 months postoperative.

### 3.5. Local Recurrence and Metastasis

No Local Recurrence Was Observed

Metastasis occurred in four patients after a mean of 14.7 months (range 2–34). All these patients died of disease after a mean of 38.5 months (range 13–71).

**Table 2.** Surgical details, complications, oncological and mechanical outcomes for patients included in the study.

| Patient # | ST (min) | RL (cm) | Contaminated RM | Hemi Resection of the Ipsilateral Distal Tibia | Tibiofibular Syndesmosis Resection | Peroneal Nerve Resection | RPM (cm) | RT | BU (Months) | CO | AS | MSTS Score |
|---|---|---|---|---|---|---|---|---|---|---|---|---|
| 1 | 133 | 7.5 | No | Yes | Yes | No | 0 | TTA | 3 | - | No | 27 |
| 2 | 100 | 11 | No | Yes | Yes | No | 0 | TTA | 4 | Infection | Revision with removal of material | - |
| 3 | 180 | 11.9 | Yes | Yes | Yes | No | 0 | TTA | 12 | - | - | - |
| 4 | 120 | 9.5 | Yes | Yes (periosteal) | No | No | 2 | PedR VFT | 20 | Peroneal nerve palsy Chronic pain | No | 30 |
| 5 | 95 | 11.1 | No | Yes (periosteal) | Yes | No | 0 | FT | 12 | Chronic pain | No | 29 |
| 6 | 118 | 16.5 | Yes | No | Yes | No | 2 | FT | 12 | - | No | 21 |
| 7 | 80 | 4 | No | No | Yes | No | 2 | IA | 3 | - | No | 28 |
| 8 | 115 | 17.8 | No | No | No | Yes | 3 | No | - | Peroneal nerve palsy Chronic pain | No | 27 |
| 9 | 120 | 18 | Yes | No | No | No | 3 | No | - | - | No | - |
| 10 | 73 | 8.5 | No | No | Yes | No | 3 | No | - | Flatfoot | Partial epiphysiodesis Achilles Tendon lengthening | 25 |

ST = Surgery time, RL = Resection length, RM = resection margins, RPM = Residual peroneal malleolus, RT = Reconstruction Type, BU = Bone union, CO = Complications, AS = Additional surgery, TTA = Tibiotarsal arthrodesis, PedR VFT = Pedicled rotational Vascularized Fibula Transposition, FT = Fibula Transposition, IA = Intercalary Allograft.

## 4. Discussion

Primary bone sarcomas of the DF are very rare. The goal of surgery is to achieve negative resection margins with the reconstruction of a stable ankle [5,7]. It is widely accepted that inadequate margins may be responsible for local recurrence and poor outcomes [21]; however, most authors described no local recurrence after resection of the DF [11,22–24]. In the case of osteosarcomas, some authors even challenged an intentional marginal resection and they reported no differences regarding the survival and the local relapse [25,26]. In our series, four patients had marginal resections, but they did not develop local recurrence.

When the resection is at least 15–20 mm above the distal tibiofibular articulation [7] or 5 mm above the growth plate in skeletally immature patients [8], the lateral malleolus may be preserved. If this is not possible because of oncological reasons, reconstruction is generally recommended. Several reconstruction techniques are available, and most of the previous series reported good to excellent results (Table 3). However, most reports in the literature are sparse case reports or small case series.

Surgical techniques promoting the immediate stability of the ankle, such as arthrodesis, were described [11,23]. Dieckmann [11] reported the largest series of crossed screws AA (five cases) and retrograde nail AA (four cases). The mean MSTS Score was comparable between the two groups [11]. In the retrograde nail AA group, even if plantar fascia irritation led to the removal of the material in one case, an excellent bone union was achieved [11]. In the case of crossed screws AA, one patient eventually underwent below-the-knee amputation because of an infected pseudarthrosis [11]. Ozaki [23] used the same technique in one case with no complications and the patient had a good functional result after a 12 cm resection of the distal fibula. In our series, infection was the only complication that occurred in these patients, and it was observed only in one patient 49 days after surgery. This patient required two surgical debridements without removal of the hardware.

**Table 3.** Previous studies in literature describe several reconstruction techniques in the treatment of distal fibula sarcomas, complications, follow-up and functional outcomes.

| RT | N. of Cases | Author | Complications | MFU (Months) | FO |
|---|---|---|---|---|---|
| AA | 9 | Dieckmann [11] | DWH, Pseudoarthrosis, Talipes equinus, Deep infection, Fracture | 39.9 | E |
| | 1 | Ozaki [23] | - | 50 | G |
| Ped VFT | 2 | Capanna [7] | Reduced ankle mobility | 15 | - |
| | 1 | De Gauzy [9] | - | 30 | E |
| NVFT | 1 | Dieckmann [11] | Intralesional resection (Amputation) | 155 | - |
| | 1 | Leibner [5] | - | 60 | E |
| | 1 | Ozaki [23] | Intralesional resection (Amputation) | 43 | P |
| BC + NVG | 1 | Ozaki [23] | Fatigue Fracture, LLD | 114 | G |
| IG | 3 | Capanna [7] | - | 12 | E |
| | 4 | Jamshidi [12] | Valgus deformity, Syndesmosis screw breakage | 38 | E |
| NR | 1 | Jung [6] | Severe valgus deformity, Limited mobilities, LLD | 84 | - |
| | 3 | Ozaki [23] | Wound Necrosis, Drop foot, Intralesional resection (Amputation) | 31 | P |
| | 3 | Capanna [7] | - | 13.5 | E |

RT = Resection Type, N. of cases = Number of Cases, MFU = Mean Follow-up, FO = Functional Outcome, AA = Ankle Arthrodesis, Ped VFT = Pedicled Vascularized Fibula Transposition, NVFT = Non-Vascularized Fibula Transposition, BC + NVG = Bone Cement and Non-Vascularized graft, IG = Intercalary Graft, NR = No reconstruction, DWH = Delayed Wound Healing, LLD = Limb Length Discrepancy, E = Excellent, G = Good, P = Poor.

Capanna [7] was the first to describe reconstruction of the DF using a pedicled vascularized fibula. In both patients reported, a stable and painless ankle was achieved. However, in one case, the functional outcome was altered because of a reduced ankle mobility. De Gauzy [9] recommended the reconstruction of the DF with pedicled vascularized fibula but only in skeletally immature patients. A stable ankle without valgus deformity and an excellent bone union was achieved [9]. In our series, this technique was used in a skeletally

mature patient (27 years old) and no mechanical complications were observed. However, this patient developed peroneal external nerve palsy during radiation therapy, but he recovered eventually. Adjuvant therapy can induce peroneal mononeuropathy, but this usually completely resolves after finishing the treatment [27,28].

Reconstructions with non-vascularized fibular grafts have generally good results [5,7,11,23]. Ozaki [23] used the contralateral fibula to reconstruct the bone defect. Even though a fatigue fracture occurred and the patient had a 1.5 cm limb length discrepancy, a good bone consolidation was observed. Dieckmann [11] and Leibner [5] described fibular transpositions in one patient, each. Patients had a stable ankle with normal ankle mobilities. In addition, when harvesting the proximal fibula, it is very important to keep in site the fibula head and the collateral knee ligament so as not to alter the knee stability [11].

Following the DF resections, allografts can be used to reconstruct the bone defect [7,12,24]. Jamshidi [12] reported four cases with an excellent MSTS Score, but in one case ankle valgus deformity occurred. Sometimes, patients may require additional complex surgeries to correct the valgus deformity [6]. To prevent this, any time the ankle mortise is part of the wide resection, Capanna suggests to restore the distal tibiofibular syndesmosis by fixing the graft to the distal tibia with a screw [7]. In our series, we used a 4 cm allograft strut fixed to the distal tibia and the patient had an excellent functional outcome without any biomechanical disturbances. Similar results, but with a shorter follow-up, were reported by San Julian [24].

In the case of preserving the lateral malleolus, DF resections might be followed by no reconstruction [7,8,23]. Ozaki [23] reported three patients with a functional score varying from poor to good. Two patients had additional surgeries for wound necrosis and amputation for intralesional resection. Capanna [7] described three patients with an excellent MSTS Score and no complications. However, he recommends syndesmosis screw fixation to prevent the lateral malleolus to rotate in valgus any time when the resection is conducted 15–20 mm above the distal tibiofibular articulation. In children, neither reconstruction nor syndesmosis fixation is required when subperiosteal resections are at least 0.5 cm above the growth plate [8].

This study has several limitations to be considered. First, the study was retrospective and, therefore, it was subject to inherent limitations and biases. The technique of reconstruction was not randomized, and the preference of the surgeon may have contributed to a selection bias. Second, it may not be possible to judge the true incidence of complications due to the limited sample size. In addition, many potentially uncontrolled variables existed, such as the amount of soft tissue excision and the characteristics of fixation. However, primary bone tumours of the DF are very rare and, to the best of our knowledge, we report one of the largest studies.

## 5. Conclusions

Due to the low incidence of DF sarcomas and the role of this bone in the ankle biomechanics, the choice of a specific reconstructive procedure is frequently based on the surgeon's preference and experience, reflecting the lack of large studies.

With the present case series, we tried to make a valid contribution to a better definition of the surgical treatment. As described, all methods have different advantages and disadvantages. Based on our series, in the case of malignant bone tumours, whenever the peroneal malleolus and lateral cortex of the distal tibia are sacrificed, we suggest AA. In young patients or, in case of a residual peroneal malleolus less than 3 cm, we preferred a fibular transposition or an intercalary allograft. For residual lengths of more than 3 cm, the DF was fixed to the tibia with a screw.

A larger series of patients with long-time follow-up is required to assess the durability of the reconstruction.

**Author Contributions:** Conceptualization, A.A.K.; methodology, A.S.; validation, D.M.D. and M.D.P.; data curation, R.Z. and C.G.; writing—original draft preparation, A.A.K.; writing—review and

editing, A.A.K. and A.S.; supervision, M.D.P. and D.M.D. All authors have read and agreed to the published version of the manuscript.

**Funding:** The Article Processing Charge was partially funded by Lenval University Children's Hospital of Nice, France.

**Institutional Review Board Statement:** The study was conducted according to the guidelines of the Declaration of Helsinki, and approved by the local Ethics Committee of the Rizzoli Orthopaedic Institute of Bologna (Protocol number: 0003067, Approval Date: 10 February 2016).

**Informed Consent Statement:** Patient consent was waived due to the retrospective nature of the study.

**Data Availability Statement:** On request from the corresponding author, the data are not publicly available due to privacy and ethical reasons.

**Acknowledgments:** The authors thank all medical teams around world who continue to fight cancer during COVID pandemics.

**Conflicts of Interest:** The authors declare no conflict of interest.

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
