# Peer review of "Distal Fibula Reconstruction in Primary Malignant Tumours"

_curroncol, doi:10.3390/curroncol28050299_

Round 1

Reviewer 1 Report

The issue is very interesting and the case series is well describe. 

Line 185 are reported intralesional margins but at line 210 are reported marginal, please clarify it.

Reviewer 2 Report

The authors Khal and colleagues investigated the restoration of ankle biomechanics after distal fibula (DF) resection bone sarcomas with different techniques including tibiotalar arthrodesis, residual lateral malleolus fixed to tibia, non-vascularized or rotational vascularized fibula transposition and intercalary allograft. In particular a retrospective analysis of 10 patients was carried out. The results showed a median follow up of 54 months, 6 patients were free of disease while 4 patients died of disease. All patients had a stable ankle and the bone union was achieved after a mean of 9.4 months. The authors conclude that after DF resection, a restoration of ankle biomechanics gave acceptable functional results, although larger case series are needed with longer follow up in order to confirm the durability of the reconstruction.

This is an interesting work and could represents a starting point for further investigations.

The manuscript would benefit from the following:

  1. In the histopathological characterization section, diagnostic features should be implemented. Representative images of hematoxylin and eosin staining should be added.
  2. The authors should provide more information about the chemotherapy and the dosage of radiotherapy administered.
  3. The authors should better explain the surgical margins status according to Enneking stage
  4. The authors should provide more information about the surgical approach
  5. Some relevant papers are missing. The following references should be added to the manuscript, in particular: “Distal fibular excision: A review of the literature and presentation of our reconstruction technique case series”. Int J Surg Case Rep. 2021 Mar;80:105611. doi: 10.1016/j.ijscr.2021.01.105. Epub 2021 Feb 11. PMID: 33621730; PMCID: PMC7905352. “Reconstructive Challenges of Distal Tibia Bone Tumors: Extracorporeally Irradiated Autograft Combined with a Nonvascularized Autograft Fibula for Superior Reconstruction and Functional Outcomes When Compared to Ipsilateral Pedicled Fibula Transfer Alone”. Sarcoma. 2021 Mar 23;2021:6624550. doi: 10.1155/2021/6624550. PMID: 33814963; PMCID: PMC8012118. “The potential role of the extracellular matrix in the activity of trabectedin in UPS and L-sarcoma: evidences from a patient-derived primary culture case series in tridimensional and zebrafish models”. J Exp Clin Cancer Res. 2021 May 11;40(1):165. doi: 10.1186/s13046-021-01963-1. PMID: 33975637. This translational work investigate the use of tridimensional culture in the study of sarcoma biology and could be promising device for regenerative medicine approach.
  6. English spell and grammar check
  7. Limitations of the study should be included

Minor revisions are requested

Author Response

Please see atachment.

Reviewer 3 Report

In this manuscript, Khal and colleagues performed a retrospective study on 10 bone sarcoma patients after distal fibula resection. The effect of various surgical techniques on the resulting ankle biomechanics characteristics was reported and discussed. The patient population, timeline of the study, individual surgical procedures, and follow-up results were clearly presented. The authors were able to generate suggestions on surgical procedures based on the current analysis. This is a good manuscript covering a rare tumour population and providing valuable updates on the surgical techniques and outcomes. Therefore, this manuscript can be considered for acceptance in its present form. 

Author Response

Please see atachment.
